# Quantifying Asymmetric Gait Pattern Changes Using a Hidden Markov Model Similarity Measure (HMM-SM) on Inertial Sensor Signals

**DOI:** 10.3390/s24196431

**Published:** 2024-10-04

**Authors:** Gabriel Ng, Aliaa Gouda, Jan Andrysek

**Affiliations:** 1Institute of Biomedical Engineering, University of Toronto, Toronto, ON M5S 1A1, Canada; gabrielp.ng@mail.utoronto.ca (G.N.); aliaa.gouda@mail.utoronto.ca (A.G.); 2Research Institute, Holland Bloorview Kids Rehabilitation Hospital, Toronto, ON M4G 1R8, Canada

**Keywords:** gait assessment, gait disabilities, machine learning, unsupervised learning, wearable sensors

## Abstract

Wearable gait analysis systems using inertial sensors offer the potential for easy-to-use gait assessment in lab and free-living environments. This can enable objective long-term monitoring and decision making for individuals with gait disabilities. This study explores a novel approach that applies a hidden Markov model-based similarity measure (HMM-SM) to assess changes in gait patterns based on the gyroscope and accelerometer signals from just one or two inertial sensors. Eleven able-bodied individuals were equipped with a system which perturbed gait patterns by manipulating stance-time symmetry. Inertial sensor data were collected from various locations on the lower body to train hidden Markov models. The HMM-SM was evaluated to determine whether it corresponded to changes in gait as individuals deviated from their baseline, and whether it could provide a reliable measure of gait similarity. The HMM-SM showed consistent changes in accordance with stance-time symmetry in the following sensor configurations: pelvis, combined upper leg signals, and combined lower leg signals. Additionally, the HMM-SM demonstrated good reliability for the combined upper leg signals (ICC = 0.803) and lower leg signals (ICC = 0.795). These findings provide preliminary evidence that the HMM-SM could be useful in assessing changes in overall gait patterns. This could enable the development of compact, wearable systems for unsupervised gait assessment, without the requirement to pre-identify and measure a set of gait parameters.

## 1. Introduction

Disabilities, such as lower-limb amputation, Parkinson’s disease, cerebral palsy, and others, significantly impact gait, leading to abnormal gait patterns [1,2,3]. These gait deviations can lead to problems such as increased energy expenditure during walking, heightened risk for falls, injuries, pain, and reduced quality of life [1,4]. Objective gait analysis and monitoring methods can improve our ability to track changes in gait patterns over time. This can better inform the rehabilitation process and enhance treatment and decision making for individuals with disabilities [5,6].

In particular, wearable systems based on inertial sensors offer the potential for cost-effective, easy-to-use, and versatile gait assessment within the lab and in free-living or community environments [5,7]. Inertial sensors could also allow for integration with smart textiles [8] and multi-modal systems [9,10] to develop comprehensive gait analysis systems. While commercial systems like the Xsens MVN can measure a wide range of spatiotemporal and kinematic variables to assess overall gait patterns [11], recent literature has highlighted the need for simpler and more usable wearable systems consisting of fewer inertial sensors that would facilitate integration into routine clinical and community-based care [7,12]. However, existing systems using just one or two inertial sensors are typically restricted to simple activity monitoring or the measuring of a reduced set of gait parameters [12]. This limits their potential to monitor changes in overall gait patterns, especially when applied to diverse clinical populations with varying gait patterns and deviations [7]. As such, rather than attempting to derive a multitude of gait parameters from one or two inertial sensors, alternative approaches have been investigated that use machine learning (ML) to assess changes in gait patterns by directly assessing the time-series signal outputs of wearable sensors (e.g., accelerometer, gyroscope, etc.) [13,14,15]. This approach eliminates the need to predefine and accurately measure or estimate many parameters of interest, enabling the design of compact, non-intrusive systems for the generalized monitoring of gait patterns.

Studies using wearable sensor signals to determine differences or changes in gait have largely used ML. This includes population-level models such as for classification of Parkinson’s disease vs. healthy gait [13,16] and healthy vs. simulated abnormal gait patterns [15,17]. Studies have also investigated classification of changes in gait patterns within the same individual, such as those due to fatigue [18], gait training [14], or exercise intervention [19], using data from one to three inertial sensors. Some studies on Parkinson’s disease have attempted to go beyond simple nominal classification and assess changes in overall gait patterns in a more continuous or ordinal manner [13,20,21]. Instead of solely determining if gait patterns are different, these models can determine the degree of change and whether gait is worsening or improving (either discretely or continuously, as in regression). These include a multiclass ML model to separate individuals into Hoehn and Yahr levels of Parkinson’s disease [20] as well as several regression models that predict unified Parkinson’s disease rating scale (UPDRS) scores [13,21,22]. However, one significant challenge is that these machine learning models require supervised training using reference external gait pattern data or physical/motor function measure (e.g., UPDRS in the studies above); many of these measures have not been shown to be adequately sensitive or responsive [23,24]. Additionally, fully supervised learning has other significant limitations when applied to real-world data and healthcare applications. This includes the need for large datasets with corresponding labels for training, poor generalization, and difficulty adapting to data that do not fall within the training distribution [25]. These challenges are especially relevant when applying ML to rehabilitation [9].

The application of unsupervised or minimally supervised frameworks to wearable sensor gait data could address the aforementioned challenges, reducing the need for large datasets and training and leading to more robust gait analysis methods. Recently, Sahraeian and Yoon have proposed a computationally low-cost framework to compare the emissions matrices of different hidden Markov models (HMM) and output a bounded measure (between 0 and 1) quantifying the similarity of two HMMs [26]. Both HMM training and the similarity comparison are unsupervised. Leveraging HMMs also offers advantages over other methods of comparing time-series data, including learning higher-level structures and system dynamics [27]. This can allow for further analysis of gait phases and temporal dynamics of gait [28,29]. Similarity measures for comparing HMMs have recently been applied in areas such as speech and text recognition as well as computational biology [30,31,32], but they have not yet been applied to or explored for use in gait analysis. Additionally, the properties of the method proposed by Sahraeian and Yoon have only been validated using artificially constructed HMM matrices, and the method has not been adapted and tested to HMMs trained on real-world data, such as gait data from inertial sensors. 

The overarching goal of this work was to explore if an HMM-based similarity measure (HMM-SM) could be used to monitor changes in gait patterns by assessing the similarity of HMMs trained on inertial sensor data (accelerometer and gyroscope time-series data) collected during gait. This work presents a preliminary examination of the HMM-SM in a controlled scenario in order to assess different degrees of changes in gait patterns. In this study, we manually perturbed gait patterns in able-bodied individuals using a previously validated biofeedback system [33]. Our objectives were to assess whether, when applied to inertial sensor data from one or two sensors, (1) the HMM-SM similarity decreased as an individual’s gait moved away from baseline gait patterns and whether (2) the HMM-SM method could provide a reliable measure of gait similarity. As part of both objectives, we also wanted to investigate the effects of HMM hyperparameters (primarily number of states) on HMM-SM performance and to identify the optimal sensor location(s) along the lower body. This work is a preliminary step toward the development of a gait analysis tool for unsupervised gait monitoring based on inertial sensor signals.

## 2. Methodology

### 2.1. Participants

As the HMM-SM has not previously been applied to gait analysis, we opted to validate its response to able-bodied (AB) walking patterns before moving to specific populations with disabilities. Eleven AB participants completed the study protocol (5 males, 6 females; age 24.09 ± 3.47 years, 1.73 ± 0.09 m, 63.64 ± 10.31 kg). AB participants were defined as individuals exhibiting no obvious gait abnormalities and having no previous history of musculoskeletal, neurological, or cardiovascular disorders. Informed consent was obtained from each participant at the beginning of the data collection session. The recruitment and the experimental procedure were each approved by the Research Ethics Board at Holland Bloorview Kids Rehabilitation Hospital (REB-0176). 

### 2.2. Data Collection Protocol

Participants completed all trials by walking back and forth alongside a 15 m-long straight pathway, turning in a tight radius (1–2 m) at either end to form a circuit. Participants began with a baseline set of walking trials consisting of five laps of the circuit. Following, each participant completed a series of trials involving rhythmic auditory stimulation (RAS) to gradually perturb their gait and produce a range of gait cycle stance-time symmetry ratios (STSR). These involved seven total trials, outlined in detail in subsequent paragraphs. Stance time is defined as the time from heel-strike to toe-off, and STSR is the ratio between right and left stance times. The equation for determining STSR is shown in Equation (1), where STSR of 1.0 indicates perfect symmetry.
(1)$$STSR=\frac{{\left(Stance\;Time\right)}_{right}}{(Stance\;Tim{e)}_{left}}$$

STSR has been demonstrated to significantly impact gait patterns [34,35] and is one of the most common gait deviations in disability populations, such as those associated with Parkinson’s disease, lower-limb amputation, cerebral palsy, and stroke [36,37,38,39]. The RAS trials allowed us to simulate a range of gait cycle STSRs and evaluate the model’s response to a controlled set of gait changes. For applying RAS, we used a system developed by our lab and described in detail by Gouda and Andrysek [33], in which participants were instructed to match their foot strikes as closely as possible to auditory beats provided via headphones. These beats were provided at the participant’s self-selected cadence, as measured during baseline walking. The system reported STSR results for each gait cycle in real time to a laptop to monitor performance for each trial. The STSR range, speed, and cadence for each participant are shown in Table 1. 

Following the baseline walking, participants completed a series of training trials while receiving RAS at a constant symmetry level for the trial duration, in order to familiarize themselves with the system’s auditory metronome. Participants trained at three levels of STSR, completing three laps of the circuit at each level: 1.0 (i.e., symmetry, to introduce participants to the auditory cueing system), 0.9, and 0.8. Participants then completed four trials where target symmetry changed throughout the trial and with each trial lasting between 5–7 min. Participants began these trials with the RAS system providing beats at an initial target symmetry, and as the trial progressed, the symmetry of the metronome gradually changed until reaching a final value. The initial and final target symmetries for trial 1 were 1.0 and 0.8, respectively. For trials 2–4, the start and end target symmetries were varied within the ranges [1.0, 0.9] and [0.8, 0.7], respectively. The symmetry targets were tailored for trials 2–4 depending on how participants responded to the beats from the previous trials, in order to produce a relatively even spread of STSR values achieved during the session. This was primarily to reduce biasing of the data toward symmetry and because some participants needed higher asymmetry beats to perturb their walking from baseline symmetry. This resulted in 323 to 544 steady-state gait cycles for each participant across all of the trials. 

### 2.3. Instrumentation and Pre-Processing

Participants were equipped with two systems. (1) A wearable motion capture system, Xsens Awinda (Xsens Technologies BV, Enschede, The Netherlands), consisting of eight inertial sensors, was attached along the lower body and sternum. This system was used as the gold-standard reference for identifying gait events and calculating gait symmetry. (2) The RAS system consisted of five Xsens DOT sensors (Xsens North America Inc., El Segundo, CA, USA) attached along the lower legs, upper legs, and pelvis, and connected to a smartphone via Bluetooth. The smartphone was worn at the pelvis, receiving the streamed inertial sensor signals and controlling the auditory stimulus. The sensor setup is illustrated in Figure 1. The inertial sensor data from the DOT sensors were used to train the ML models. The Xsens DOT inertial sensors collected raw tri-axial gyroscope (range ±2000°/s) and accelerometer (range ±16 g) data at 60 Hz which were then passed through a lowpass Butterworth filter at 40 Hz. Heel-strike and toe-off gait events were identified from the Xsens Awinda foot contact information, which were then time-aligned to the DOT sensor data to segment the data into gait cycles for each of the five sensors. Gait cycles occurring during turns (i.e., at either end of the circuit) were excluded from the analysis.

### 2.4. HMM-SM Model Implementation

To investigate the use of ML for the assessment of gait similarity, our method involved two main steps, as follows:

Given sets ${SL}_{1},\ldots ,{SL}_{L}$, where each set $SL_{i}$ (i∈1,…,L) represents gait around a certain mean stance-time symmetry level, train HMMs $\lambda_{1},\ldots ,\lambda_{L}$ for each respective set. The input to each model is made up of time-series data obtained directly from the inertial sensors.Using a measure for comparing the similarity of HMM models
$S(\lambda_{a}||\;\lambda_{b})$, evaluate whether the similarity measure appropriately responds to differences between ${SL}_{a}\;and\;{SL}_{b}$ (detailed explanation in Section 2.5).

An HMM is a probability-based model which characterizes a system as a Markov process capable of transitioning between *N* states [40]. The “hidden” aspect refers to the fact that the true states of the system are not directly observed or recorded, but rather are inferred through an external measurement, such as accelerometer and gyroscope data. This inference is made in an unsupervised manner. Each HMM is characterized by two main matrices, *A* and *B*. *A* is an *N* × *N* matrix where each element $a_{i,\;j}$ represents the probability the system would transition from state *i* to state *j* at any given time. *B* is the emissions probability matrix consisting of *N* elements $b_{1},\ldots ,b_{N}\;$ where each element represents the likelihood of system outputs for a given state [40]. These likelihoods are represented by a μ (mean) and covariance matrix. 

The advantage of HMMs is that the model parameters *A* and *B* represent probability distributions and offer an easily interpretable representation of a system’s underlying structure [41]. This contrasts with the relatively black-box approach of many ML and deep learning methods, such as neural networks and support vector machines (SVM), which can be difficult to interpret [42]. It is possible to directly compare the model parameters of $\lambda_{a}$ and $\lambda_{b}$, rather than needing to use an external task and performance metric, such as classification accuracy or regression mean squared error. To compare the HMM parameters, we applied a similarity measure from Sahraeian and Yoon [26], referred to in this work as the HMM-SM. To determine the similarity between $\lambda_{a}\;and\;\lambda_{b}$, we first calculated a state correspondence matrix *Q* where each element $q_{i,j}$ indicates the overall similarity between the emission matrix elements $b_{a}[i]\;and\;b_{b}[j]$, evaluated using symmetric Kullback–Leibler (KL) divergence. Subsequently, a similarity measure $S(\lambda_{a}||\lambda_{b})$ was calculated based on the sparsity of *Q*, as follows:(2)$$S(\lambda_{a}||\lambda_{b})≜\frac{1}{2}\left[\frac{1}{M}\sum_{i=1}^{M}H\left(r_{i}\right)+\frac{1}{M^{\prime }}\sum_{i=1}^{M^{\prime }}H\left(c_{j}\right)\right]$$
where *r_i_* is the *i-*th row of *Q*, *c_j_* is the *j-*th column of *Q*, and *M* and $M^{\prime }$ are the number of rows and columns in *Q*, respectively. H(u) represents the normalized Gini index, which returns the sparsity of vector ***u*** from 0 to 1. This is based on the intuition that, for two similar HMMs, each state in $\lambda_{a}$ will be highly similar to one state in $\lambda_{b}$ (high value in *Q*) and relatively dissimilar to the other states (low value in *Q*). Thus, a sparser *Q* should indicate higher similarity. More dissimilar HMMs will have a more even distribution of numbers throughout *Q* and subsequently a lower sparsity, because for each state in $\lambda_{a}$, it is less likely for one state in $\lambda_{b}$ to be more similar than the rest. The result of this sparsity analysis is a single output representing the similarity between two HMMs, $\lambda_{a}\;$ and $\lambda_{b}$, where output closer to 1 (higher sparsity) indicates a greater similarity of datasets. 

### 2.5. Model Training and Analysis

HMM training and HMM-SM evaluation were conducted on a subject-specific basis, using the accelerometer and gyroscope signals from the gait cycles extracted for each participant as the input data to train the HMMs. The overall process is illustrated in Figure 2. Model implementation and data analysis were implemented using Python 3.8. Statistical tests used the open-source Pingouin library [43] with HMM implementation based on the open source hmmlearn library [44]. HMMs were fit to each set of gait cycle data, SL, using an expectation–maximization algorithm. The transition matrix *A* was restricted to a left-to-right architecture based on prior knowledge of gait [45]. The covariance matrix *B* was fully populated (i.e., not restricted to non-zero variance only on the diagonal) and fit to Gaussian emissions. Our work aimed to evaluate the similarity of the trained HMM models, as opposed to the response of the HMM models to unseen gait data. For this reason, data were not split into training and test as in supervised classification or regression analyses. In other words, HMMs were trained on all data within the respective set, SL. 

Two paradigms were used to evaluate the response of the model corresponding to objectives 1 and 2. For both methods, gait data were separated into sets representing different symmetry levels ($SL_{1}$,…, $SL_{L}$). Section 2.5.1 and Section 2.5.2 explain how the symmetry levels were formed for each paradigm as well as their motivations, and Table 2 shows the symmetry ranges and total number of gait cycles per participant under the paradigms. 

Preliminary experiments demonstrated that concatenating gait cycles increased HMM training consistency, with no significant changes in training when using more than four gait cycles. For training the models we transformed the data within each set to form multi-gait cycle sequences. This was achieved by randomly shuffling the gait cycles in each set, ${SL}_{i}$, then iteratively concatenating groups of four gait cycles along the time axis. Ten HMMs were trained for each symmetry level, with the order of sequences shuffled between training sessions. We then compared all permutations of the trained HMMs using the similarity measure described in Section 2.4. This allowed us to calculate a mean HMM-SM similarity between each of the symmetry levels for each participant, ${\bar{S}(SL}_{a}||\;{SL}_{b})$. ${\bar{S}(SL}_{a}||\;{SL}_{b})$ denotes a mean similarity comparing all permutations of HMMs in ${SL}_{a}$ and ${SL}_{b}$. This was calculated both between symmetry levels and within the same symmetry level, as shown in Figure 2.

We explored the following sensor configurations and their impact on HMM-SM performance: pelvis, upper left leg (UL), upper right leg (UR), lower left leg (LL), lower right leg (LR), as well as the combined UL + UR signals and combined LL + LR signals. The single-sensor data were 6 × T arrays (tri-axial gyroscope and accelerometer signals, over T time points), whereas for the combined signals the sensor data were stacked such that the HMMs were training on 12 × T arrays.

Most gait classification and analysis studies using HMMs use six or fewer states [13,16,46]. In this study, we tested using two, three, four, and five states for the HMMs.

#### 2.5.1. HMM-SM Validity Compared with STSR (Three-Symmetry-Level Paradigm)

For this paradigm, each participant’s gait cycle data were split into three levels, as follows: $SL_{1}$, $SL_{2}$, and $SL_{3}$. This was done to test whether the HMM-SM decreased as STSR moved away from a baseline symmetry level. Using STSR as the ground truth to measure changes in gait patterns, we would expect to see that the similarity output decreases for each participant as their STSR level moves further away from a baseline symmetry level; for example, $\bar{S}({SL}_{1}||{SL}_{1})$ > ${\bar{S}(SL}_{1}||\;{SL}_{2})$ > ${\bar{S}(SL}_{1}||\;{SL}_{3})$. The STSR ranges were determined such that gait cycle data were split approximately evenly across the symmetry levels, that none of the clusters overlapped, and that a gap of at least 0.01 was maintained between adjacent clusters. For example, if the minimum symmetry for $SL_{2}$ was 0.95, the maximum symmetry for $SL_{3}$ would be 0.94. Though literature is sparse in establishing a minimum detectable change (MDC) of STSR, one study suggests it is around 3–4% for able-bodied individuals [47]. Therefore, we maintained a minimum difference of 0.03 between the mean STSR of each symmetry level. The specific symmetry values for each participant are shown in Table 2.

A statistical significance approach was chosen over a regression due to the number of data points required to train the HMMs. A Shapiro–Wilk test was used to confirm that the differences were normally distributed. Subsequently, a repeated-measures ANOVA (RM-ANOVA) test was used to determine whether the HMM-SM was significantly different for the symmetry levels within each participant. Following the RM-ANOVA, post-hoc paired *t*-tests with a Bonferroni correction for multiple comparisons were used to determine which levels differed and whether these differences aligned with the anticipated trends. The statistical package multiplied the raw *p*-values by the Bonferroni correction factor and reported the adjusted *p*-values, while maintaining the alpha level used (0.05) [48]. The values in the results and Appendix A report these adjusted *p*-values.

A visual explanation of the expected HMM-SM response comparing models trained on the various symmetry levels is shown in Figure 3. From the pairwise comparison tests, we would expect a perfect model response to achieve significant differences corresponding to the eight relationships that are highlighted by the significance bars in Figure 3.

#### 2.5.2. HMM-SM Reliability (Two-Symmetry-Level Paradigm)

For this paradigm, each participant’s data were partitioned into two symmetry levels: $SL_{1}$ and $SL_{2}$. The specific STSR ranges for each participant are also shown in Table 2. Data within each respective symmetry level were then randomly partitioned into two equal-sized sets, $SL_{X}$ and $SL_{X}$*. $SL_{X}$ and $SL_{X}$* do not share any gait cycle data but represent a similar distribution of gait STSR; these were verified to have the same mean and standard deviation. This paradigm aimed to assess the reliability of the HMM-SM method for comparing gait similarity. We would expect a reliable gait similarity measure to produce similar outputs when comparing HMM models regardless of whether $SL_{X}$ or $SL_{X}$* are interchanged, which is further explained in Figure 4. Reliability is an important characteristic for gait measures, as it ensures the method will be sensitive enough to identify relevant changes in gait patterns [49].

To assess the reliability of the HMM-SM, we computed the intraclass correlation coefficient (ICC) using a two-way mixed effects model and consistency. A higher ICC would indicate that the HMM-SM was able to provide a reliable output of gait similarity (i.e., outputs we would expect to be similar to each other are similar). For this study, we considered an ICC of less than 0.5 as poor reliability, between 0.5 and 0.75 as moderate, between 0.75 and 0.9 as good, and greater than 0.9 as excellent, based on similar existing studies with the gait profile score (GPS) [50,51]. We also calculated the standard error of measurement (SEM) and minimal detectable change (MDC) with a 95% confidence threshold [52]. In particular, the MDC provides useful information for understanding the responsiveness of a measure [52]. This was tested for the optimal sensor locations identified from the three-symmetry-level paradigm.

## 3. Results

The HMM-SM was evaluated for validity and reliability using the three-symmetry and two-symmetry-level paradigms, respectively. The results for these are shown in Table 3 and Table 4.

### 3.1. HMM-SM Validation—Three Symmetry Levels

Table 3 shows the number of significant differences based on the pairwise *t*-tests for each of the sensor locations and HMM model configurations for the three-symmetry-level paradigm. The individual *p*-values for all comparisons and configurations can be found in the Appendix A. Figure 5 shows individual results from the pelvis, UL + UR and LL + LR five-state HMM testing as an example of configurations which were able to achieve six or more significant differences. Some examples of poor sensor location/hyperparameter selection are shown in the Appendix A.

Three configurations achieved all eight significant differences when using the HMM-SM to compare the models: UL + UR (3 HMM states) and LL + LR (2 and 5 HMM states). Additionally, 17 of the 28 tested model/sensor configurations achieved six or more of the expected significant differences as seen in Table 3, with at least one configuration at each of the sensor locations (five single-sensor lower body locations and the two combined sensor signals). For 15 of those 17 configurations, ${\bar{S}(SL}_{1}||\;{SL}_{2})$ and ${\bar{S}(SL}_{1}||\;{SL}_{3})$ were less than ${\bar{S}(SL}_{1}||\;{SL}_{1})$, and for all 17, ${\bar{S}(SL}_{3}||\;{SL}_{1})$ and ${\bar{S}(SL}_{3}||\;{SL}_{2})$ were less than ${\bar{S}(SL}_{3}||\;{SL}_{3})$, which can be seen in the Appendix A.

For sensor location, the best overall result was the combined UL + UR signals (6.75 mean achieved differences across the states tested), followed by the LL + LR (6.25) and pelvis (6.00). For both the UL + UR and pelvis configurations, statistical tests on the HMM-SM response indicated at least six of the expected significant differences for all of the HMM states tested. This means that these configurations were always able to indicate changes from the baseline STSR. Both UL + UR and LL + LR outperformed all single-sensor locations, both in mean and max number of significant differences achieved. This was expected as the HMM could train on more information. However, the LL + LR configuration exhibited more variable performance, where the HMM-SM response followed all the expected relationships when using two or five states, but the three- and four-state configurations performed poorly.

None of the single-sensor leg configurations, either in the upper or lower legs, achieved above 5.50 differences on average. Within the single-leg configurations, the upper legs sensors performed better than the lower leg configurations. The UR and UL achieved respective differences of 5.00 and 5.50 on average, whereas both the lower-leg configurations achieved less than 5 for mean differences achieved. The HMM-SM was only able to achieve, on average, 4.25 and 4.50 of the expected differences for the LR and LL configurations, respectively.

When comparing the number of HMM states, applying the HMM-SM to the three-state and five-state HMMs allowed it to achieve the expected significant differences most consistently. These states had the highest mean differences achieved across the tested sensor configurations (5.86 for both). The HMM-SM with four-state HMMs performed the worst, with a mean difference of only 5.00 across the seven sensor configurations. 

### 3.2. HMM-SM Reliability—Two Symmetry Levels

Table 4 shows the ICC scores for the HMM-SM model under the two-symmetry-level paradigm, as well as the calculated SEM and MDC values. This was to assess how the HMM-SM responded to similar gait data distributions. Analysis was conducted for the sensor configurations which had the highest mean performance across the HMM states, which were the pelvis as well as the two combined-signal configurations. Both combined-sensor signals displayed good reliability, as they exceeded the 0.75 ICC threshold (0.803 for UR + UL and 0.795 for LR + LL). This demonstrates that the HMM-SM was able to provide consistent comparisons when $SL_{X}$ and $SL_{X}$* were interchanged using two sensors along the lower body. The pelvis sensor showed only moderate reliability, with an ICC of 0.594. Additionally, the pelvis MDC (0.219) was 0.85 higher than either of the combined sensor configurations and over 20% of the range of the HMM-SM (0 to 1).

## 4. Discussion

This work presents a novel application of an ML-based similarity measure, the HMM-SM, for assessing overall gait similarity based on inertial sensor signals. This differs from most existing ML gait comparison techniques, which have largely focused on classification under supervised learning conditions. Thus, those methods have limited applicability to continuously monitor and assess changing gait patterns over time. The performance assessment had two main objectives: validate that, (1) as gait STSR moved away from an individual’s baseline STSR, the HMM-SM output decreased, and that (2) the HMM-SM provided reliable outputs when presented with separate data from similar gait distributions within an individual. We also identified the optimal model hyperparameters, primarily the number of HMM states, and sensor location(s) to guide future investigation and system design. The overall results demonstrate that the HMM-SM decreased in correspondence with changes in stance-time symmetry levels, indicating that this measure could be used to assess changes in gait using inertial sensor signals.

### 4.1. HMM-SM Validity Compared with STSR—Model Analysis Using Three Symmetry Levels

The three-symmetry-level setup was used to assess whether the HMM-SM output corresponded to increasing deviation in STSR from baseline gait patterns, evaluated through the pairwise statistical comparison. The HMM-SM achieved all the differences for configurations in the UL + UR (three states) and LL + LR (two and five states). This demonstrates the potential for the HMM-SM output to monitor the degree of changes in gait patterns, as the HMM-SM trends corresponded to clinically relevant changes in gait STSR. For these configurations, the HMM-SM was able to do more than just indicate changes from baseline STSR, as it also appropriately assessed the degree/magnitude of change relative to the changes in STSR (e.g., ${\bar{S}(SL}_{1}||\;{SL}_{1})$ > ${\bar{S}(SL}_{1}||\;{SL}_{2})$ > ${\bar{S}(SL}_{1}||\;{SL}_{3})$). This would be important for assessing changes in gait patterns over time, allowing for comparison between many time points to a baseline gait pattern to assess the amount of change over time.

Additionally, the HMM-SM consistently indicated changes from baseline STSR for over half the tested configurations. Our main objective was to determine whether the HMM-SM could be used to assess varying degrees of change in gait patterns, which was achieved for the three configurations discussed previously. However, the statistical results also demonstrated that, with a single sensor placed anywhere along the lower body, the HMM-SM output could still be used to identify when STSR had changed from a baseline STSR (e.g., ${\bar{S}(SL}_{1}||\;{SL}_{1})$ compared to ${\bar{S}(SL}_{1}||\;{SL}_{2}))$. Even for configurations where the HMM-SM was not responsive enough to differentiate between levels such as ${\bar{S}(SL}_{1}||\;{SL}_{2})$ vs. ${\bar{S}(SL}_{1}||\;{SL}_{3})$, it was able to function similar to ML models— the one-class SVM outlier detection algorithm implemented by Kobsar et al. [19] in particular— for identifying person-specific changes from baseline gait [14,18,53]. In other words, with an HMM trained on baseline gait data (in our case based on STSR), the HMM-SM output consistently identified changes from baseline gait patterns (outliers with respect to baseline).

#### 4.1.1. Sensor Location Investigation

We investigated the effect of different sensor locations and the number of states for the HMM models on the HMM-SM output. The aim was to determine the optimal configuration to achieve the ideal HMM-SM response as outlined in Figure 3 and Section 2.5.1 for the three-symmetry-level paradigm. Results from the pelvis and UL + UR, as well as comparisons between the upper and lower legs for the single-sensor configurations, indicate that gait changes could be more consistently identified using inertial sensors closer to the body’s center of mass rather than the distal regions. However, the reason why the HMM-SM in our study performed more consistently for the pelvis and thigh could be that each participant may have employed different strategies to match their gait to the rhythmic stimulus. Previous literature has shown that, even given identical gait interventions, such as gait training with a mass on the paretic side, the resulting changes in the paretic and non-paretic side can vary across individuals [54,55]. In response to the feedback system employed in this study, one participant might use more of their left side to compensate for the asymmetric beat, another might use more of their right side, or a combination of both. A location such as the pelvis or upper legs might better capture changes on either side, whereas a more distal location on one side (e.g., LR) might not capture the changes well if they are primarily occurring on the opposite side. Additionally, Kang and Dingwell have demonstrated segments further up the body (e.g., trunk, pelvis) to have higher stability than lower segments (e.g., feet) [56], suggesting areas such as the pelvis and thigh might provide more consistent gait signals for analysis with the HMM-SM. This shows that it could be feasible to design future systems involving the phone or sensors embedded in a belt or waist-strap as a convenient way to monitor for changes in overall gait patterns. 

Conversely, the two combined sensor locations (UL + UR and LL + LR) could have the best discriminatory potential, as these identified the most significant differences, and both had configurations which identified all eight differences. Activity recognition studies comparing hip (pelvis), thigh, and ankle sensors report mixed results concerning which location achieves the best accuracy and performance [57], and sensor placement varies across different applications [12,58]. However, in gait event detection, the upper and lower legs are two of the most common locations [59], suggesting that these locations may be beneficial for identifying gait features relevant to assess changes in gait patterns.

#### 4.1.2. HMM State Investigation

For the three-symmetry-level paradigm, we trained HMMs with different numbers of states to determine whether this influenced the HMM-SM performance. Our results indicate that the number of states in the HMM model impacts HMM-SM performance, with three and five states leading to the best performance. The five-state HMM also achieved more than six of the ideal trends for the UL + UR, and LL + LR signals. This may indicate that training HMMs that can divide the gait cycle into finer distinctions could be useful for the HMM-SM method in monitoring changes in the gait cycle. The two-state HMM was able to effectively differentiate between the three levels of symmetry for the pelvis, as well as the combined UL + UR and LL + LR signals, but it struggled when trained on any of the individual sensor signals along the legs. This suggests that the pelvis and the combined UL + UR and LL + LR signals are more effective in capturing gait changes at each symmetry level. Conversely with only a single sensor on the leg, a two-state HMM might not provide enough detail to reliably differentiate between the changes in gait symmetry. In general, an increase in the number of HMM states allows for increased complexity in modeling the system [60], potentially explaining the worse average performance of the HMM-SM with two HMM states compared with that with three or five states. An explanation for why the four-state HMM performed poorly compared with the three- and five-state HMMs could be the presence of some variability in HMM training and parameter convergence. This could have contributed to the worse performance in the LR and LL + LR configurations. Access to larger amounts of gait cycles for training the HMMs could decrease this variability. With more consistent training, it is possible the overall performance of the four-state HMM would be closer to the three- and five-state configurations.

### 4.2. HMM-SM Reliability—Analysis Using Two Symmetry Levels

This was tested using the optimal configurations identified in Section 4.1.1, the pelvis and the combined sensor signals. Both the UR + UL and LR + LL signals displayed good reliability with respect to interchanging similar gait distributions $SL_{X}$ and $SL_{X}$*, with ICCs of 0.803 and 0.795, respectively. This is within the range of ICC values obtained by post-stroke evaluation of GPS and GPS sub-score reliability, though at the lower end [50]. This indicates our method is not overly sensitive to the training data. There is inherent variability to gait due to natural temporal fluctuation, variations in the environment, pathology in neurological or musculoskeletal control [61]. Hence, models which can discern significant changes in gait while being resistant to stride-to-stride fluctuations are critical. These results are promising, and they support the idea that the HMM-SM method could be used to provide a robust measure of overall gait pattern similarity even between gait patterns near the MDC for STSR, which is valuable for the long-term monitoring of real-world gait conditions.

On the other hand, the pelvis sensor only displayed moderate reliability, with an ICC of 0.594. This was unexpected given the consistent performance in the three-symmetry-level paradigm, but it might show that having only one sensor reduces the reliability of the model in assessing gait similarity. The MDC was over 60% higher for the pelvis than either UL + UR or LL + LR, which indicates that larger changes in the HMM-SM would need to be observed, if a sensor at the pelvis were used, before being able to conclude that gait patterns had significantly changed. This would also explain why the pelvis was not able to achieve all eight significant differences for any of the number of states tested. 

### 4.3. Future Work and Limitations

Future studies should investigate whether increasing the amount of training data for the HMMs would have an impact on the HMM-SM method. In this study, the symmetry levels in the two-symmetry-level paradigm often included a range of 6–7% in stance-time symmetry (e.g., $SL_{2}$ could include strides with STSR values ranging from 0.90 to 0.97). If we had enough data to train HMMs on a narrower range of symmetry values, such as only gait cycles within 3–4% of each other (or the relevant MDC for the corresponding clinical population), it might be possible to train more specific HMMs (i.e., with a tighter range of convergence for HMM parameters) and achieve higher ICCs for the HMM-SM comparisons. Having more gait cycle data could also lead to more consistent HMM learning and subsequently more reliable HMM-SM output [62,63].

It is also important to investigate the effects of different conditions on the HMM-SM, such as walking outside of a lab environment [64,65] or additional variability that may be present in disability gait [66,67]. The current study is limited to straight walking, so real-world applications should either investigate the impact of including turning data or incorporate algorithms to identify and exclude turns [68]. Another limitation is the use of the Xsens Awinda for gait segmentation. To support development of a self-contained system, future work should also seek to integrate gait event detection using the same sensors used for the HMM-SM [69,70,71,72] or to evaluate the HMM-SM on gait which is not segmented by gait cycles, such as undertaken in [73], where a gait segmentation approach that used fixed time windows rather than gait events was employed. Future work should also assess the reliability of the HMM-SM in response to changes in sensor orientation, as real-world application could involve repeated donning and doffing of the sensor(s), resulting in slight changes in position and orientation over time.

Additionally, the impact of parameters such as gait speed on the HMM-SM output should be investigated, to assess how or if changing the magnitude of the inertial sensor signals affects the HMM-SM performance. Furthermore, other gait parameters apart from STSR should be explored to further validate the HMM-SM’s ability for the generalized monitoring of gait patterns. A limitation of this study is its focus on asymmetric gait pattern changes. Extending the model to assess symmetric gait deviations could validate its applicability to populations with more symmetric gait deterioration such as for those with moderate stage Parkinson’s disease [36]. The HMM-SM should also be applied to gait data from relevant disability populations, as this can introduce additional gait deviations or variability [36,66,74]. 

Although the pelvis sensor was outperformed by the combined sensor configurations (UL + UR and LL + LR), it performed the best of the single-sensor configurations tested. For developing minimally intrusive systems, it may be advantageous to design algorithms suited for single-sensor setups, such as a waist-worn belt. Future work should explore alternative HMM architectures (e.g., layered HMM, autoregressive HMM, etc.) to determine if the performance using the pelvis sensor can be further improved, particularly with regard to reliability.

Lastly, this method was applied in a person-specific context and can only be used to measure changes relative to an individual’s baseline. This could be useful for monitoring gait deterioration over time or assessing retention levels following gait training to provide feedback outside of the clinic. However, future investigation using the HMM-SM to compare gait to a fixed reference point, such as able-bodied individuals or “ideal” gait for specific disability populations, could enable this method to provide a simple, absolute measure for evaluating gait patterns within a compact, wearable system. 

## 5. Conclusions

Objective methods of quantifying gait patterns can allow us to accelerate and improve the rehabilitation process for individuals with lower-limb disability. Our method was able to provide a measure of overall gait similarity which could be used to monitor changes in an individual’s gait patterns over time. In this work, we have demonstrated that our method could appropriately respond to changes in stance-time symmetry and provide a reliable measure of gait similarity. This is an important step toward the development of models capable of generalized assessment of the gait patterns and changes in gait. This work also compared the effect of different sensor locations along the lower body, and these results could be used to inform the future design of compact wearable systems embedded in smartphones or waist-worn devices. This is particularly important for incorporating methods into routine activity in order to track gait changes and guide rehabilitation decisions, as minimizing hardware requirements can facilitate the adoption of these gait analysis systems. 

## Figures and Tables

**Figure 1 sensors-24-06431-f001:**
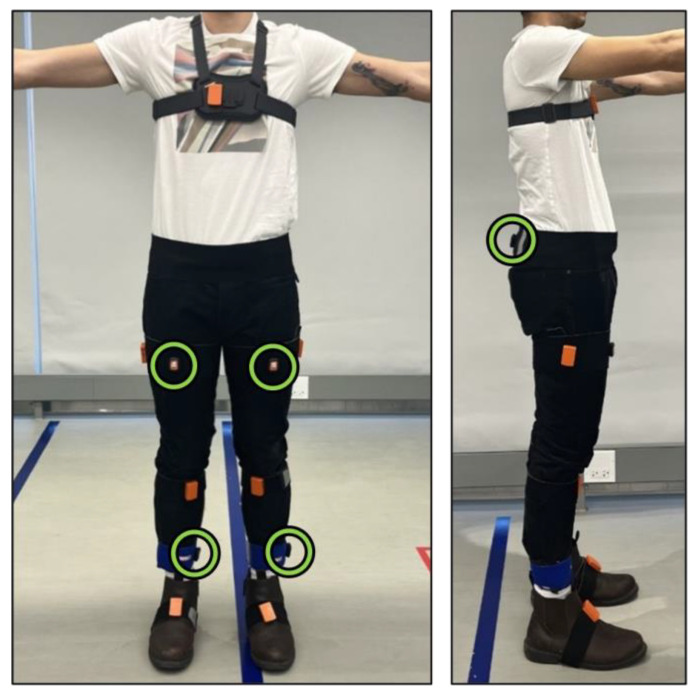
Participants were outfitted with two systems. (1) Eight Xsens Awinda sensors, the larger orange sensors, were placed as follows: two on each foot, lower leg, and thigh, as well as the back of the pelvis (underneath the waist strap) and sternum. (2) Five Xsens DOTs, indicated by green circles. These streamed tri-axial gyroscope and accelerometer data to smartphone.

**Figure 2 sensors-24-06431-f002:**
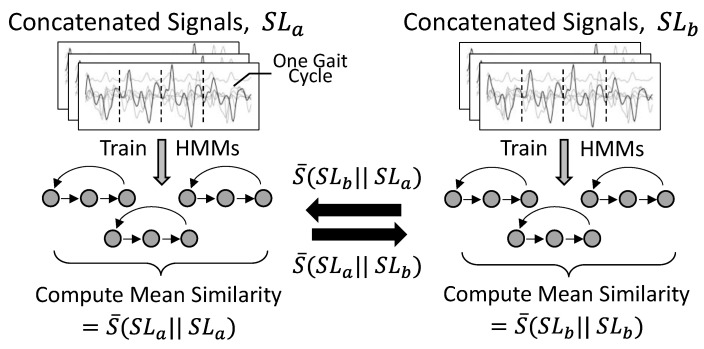
Overview of HMM-SM process. HMMs trained on concatenated signals for each symmetry level, formed by concatenating four gait cycles. Input data to the HMM were of the form M × 6 × T (or M × 12 × T for the combined signals), where M is the number of data samples in set SL following the gait cycle concatenation, 6 (12) is the number of signal axes, and T is the length of the signal. Comparing each of the HMMs, we computed average HMM-SM similarity within each level as well as between symmetry levels.

**Figure 3 sensors-24-06431-f003:**
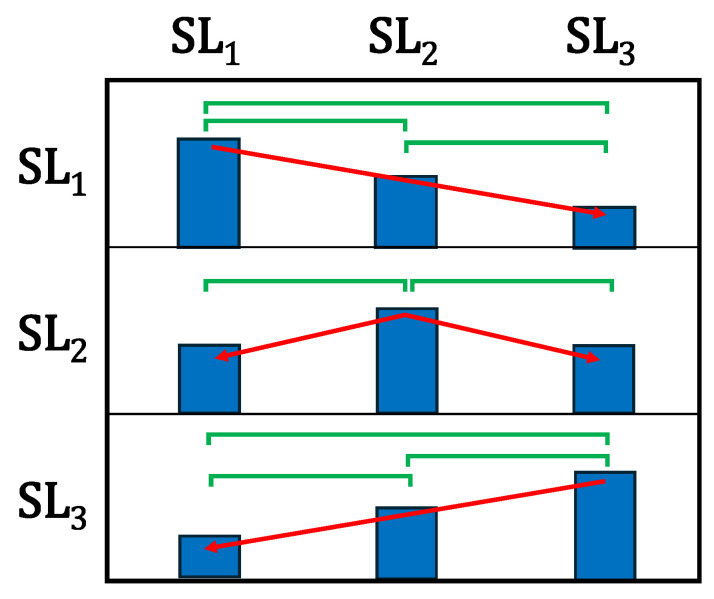
Visual explanation of HMM-SM comparisons. Vertical axis = the first symmetry level to compare and horizontal axis = the second, e.g., the top-right group corresponds to ${\bar{S}(SL}_{1}||\;{SL}_{3})$. Bars show the ideal HMM-SM response corresponding to changes in stance-time symmetry ratio. Red arrows show the expected trends for the scores, and green significance bars indicate expected differences.

**Figure 4 sensors-24-06431-f004:**
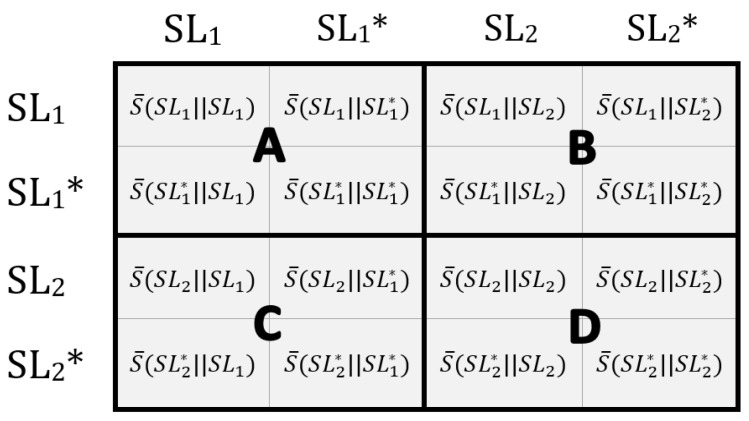
Visual explanation of the two-symmetry-level testing. HMM-SM results should be similar for model comparisons within their respective groups. For example, $\bar{S}(SL_{1}||SL_{1})$ HMM-SM output should be similar to $\bar{S}(SL_{1}||SL_{1}^{*})$ and other comparisons within group A. Additionally, we would expect outputs within groups B, C, or D to be similar to outputs in their respective groups. ICC values report the consistency of scores within groups A, B, C, and D across the participants.

**Figure 5 sensors-24-06431-f005:**
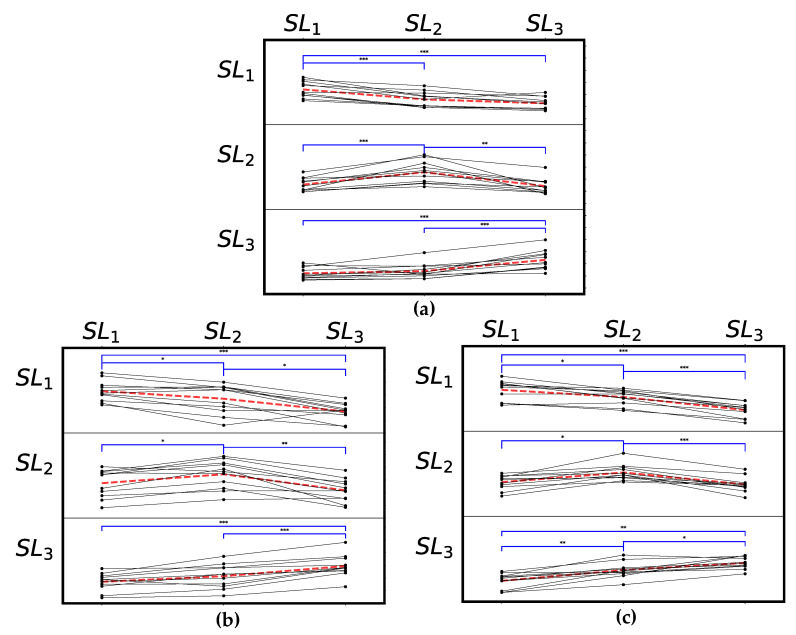
Examples of good HMM-SM performance for the 11 participants, using results from the five-state HMMs. Significance bars indicate difference between levels, determined using post-hoc paired *t*-tests. The red dashed lines indicate the mean response across participants. (**a**) Pelvis, (**b**) combined upper leg sensors, and (**c**) combined lower leg sensors.

**Table 1 sensors-24-06431-t001:** Participant gait characteristics.

Participant	STSR [Min, Max]	Cadence (Steps/min)	Speed (m/s)
1	[0.87, 1.05]	103.33 ± 3.23	1.05 ± 0.08
2	[0.90, 1.13]	112.12 ± 3.28	1.13 ± 0.15
3	[0.88, 1.07]	112.59 ± 2.88	1.24 ± 0.06
4	[0.90, 1.07]	102.60 ± 3.37	0.95 ± 0.06
5	[0.83, 1.02]	104.88 ± 2.27	1.16 ± 0.09
6	[0.83, 1.04]	108.83 ± 2.60	1.41 ± 0.09
7	[0.87, 1.02]	97.49 ± 2.73	0.83 ± 0.07
8	[0.89, 1.03]	105.95 ± 2.40	1.03 ± 0.06
9	[0.92, 1.04]	107.94 ± 3.30	0.99 ± 0.06
10	[0.88, 1.05]	103.91 ± 2.50	1.13 ± 0.05
11	[0.85, 0.99]	101.90 ± 1.98	1.27 ± 0.07

Abbreviations: STSR, stance-time symmetry ratio; min, minimum; max, maximum.

**Table 2 sensors-24-06431-t002:** Symmetry ranges for three-symmetry-level and two-symmetry-level paradigms.

	Three Symmetry Levels				Two Symmetry Levels		
Participant	Stance-Time Symmetry [Min, Max]			Number of Gait Cycles	Stance-Time Symmetry [Min, Max]		Number of Gait Cycles
	SL_1_	SL_2_	SL_3_		SL_1_	SL_2_	
1	[0.87, 0.91]	[0.94, 0.98]	[1.01, 1.05]	544	[0.90, 0.94]	[0.99, 1.05]	464
2	[0.90, 0.98]	[1.00, 1.04]	[1.07, 1.13]	405	[0.97, 1.04]	[1.06, 1.11]	461
3	[0.88, 0.95]	[0.97, 1.01]	[1.02, 1.07]	406	[0.92, 0.97]	[1.01, 1.05]	456
4	[0.90, 0.95]	[0.97, 1.01]	[1.02, 1.07]	353	[0.94, 0.98]	[1.01, 1.05]	369
5	[0.83, 0.87]	[0.90, 0.95]	[0.98, 1.02]	344	[0.85, 0.90]	[0.95, 1.01]	442
6	[0.83, 0.88]	[0.91, 0.96]	[0.99, 1.04]	435	[0.86, 0.92]	[0.96, 1.02]	499
7	[0.87, 0.93]	[0.94, 0.97]	[0.98, 1.02]	362	[0.89, 0.94]	[0.96, 1.01]	323
8	[0.89, 0.94]	[0.95, 0.98]	[0.99, 1.03]	348	[0.92, 0.96]	[0.98, 1.02]	379
9	[0.92, 0.96]	[0.97, 1.00]	[1.01, 1.04]	346	[0.94, 0.99]	[1.00, 1.06]	383
10	[0.88, 0.95]	[0.96, 0.98]	[0.99, 1.05]	373	[0.90, 0.97]	[0.98, 1.05]	416
11	[0.85, 0.90]	[0.91, 0.94]	[0.95, 0.99]	441	[0.85, 0.91]	[0.93, 1.00]	469

Abbreviations: Min, minimum; max, maximum.

**Table 3 sensors-24-06431-t003:** HMM-SM validation—significance results for three-symmetry-level paradigm.

	Number of Significant Differences out of Eight (α = 0.05, Based on Adjusted *p*-Values)				
Sensor Location	2 States	3 States	4 States	5 States	Mean (Location)
Pelvis	6	6	6	6	6.00
Upper right	4	4	6	6	5.00
Upper left	4	6	6	6	5.50
Lower right	4	6	3	4	4.25
Lower left	4	6	4	4	4.50
Upper right + upper left	6	8 *	6	7	6.75
Lower right + lower left	8 *	5	4	8 *	6.25
Mean (states)	5.14	5.86	5.00	5.86	

Number of significant differences achieved for each of the sensor and HMM state configurations tested for the HMM-SM. The ideal response shows eight significant differences. Configurations which achieved all eight are indicated with an asterisk (*).

**Table 4 sensors-24-06431-t004:** HMM-SM validation—reliability results for two-symmetry-level paradigm.

Sensor Location	ICC	SEM	MDC	ICC Confidence Interval (95%)
Pelvis	0.594	0.079	0.219	[0.43, 0.74]
Upper right + upper left	0.803	0.048	0.134	[0.70, 0.89]
Lower right + lower left	0.795	0.048	0.134	[0.69, 0.88]

Abbreviations: ICC, intraclass correlation coefficient; SEM, standard error of measurement; MDC, minimal detectable change.

## Data Availability

The data presented in this study are available on request from the corresponding author. The data are not publicly available due to ethics restrictions on data dissemination and storage.

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
