# Peer review of "Quantifying Asymmetric Gait Pattern Changes Using a Hidden Markov Model Similarity Measure (HMM-SM) on Inertial Sensor Signals"

_sensors, 2024, doi:10.3390/s24196431_

Round 1

Reviewer 1 Report

Comments and Suggestions for Authors

Dear authors,

Thank you for giving me the opportunity to review your work titled: Quantifying Asymmetric Gait Pattern Changes using a Hidden Markov Model Similarity Measure (HMM-SM) on Inertial Sensor Signals.

After reviewing your work, I believe it requires significant improvements both in methodology and presentation. Below are some comments:

Template Review

Template: The manuscript does not seem to comply with the journal’s publication guidelines.

Line 30: Why is the word “DISABILITIES” in uppercase?

Line 31: Consecutive citations with more than two references should be formatted as [1-3].

Line 31: The sentence “These gait deviations can lead to problems such as increased energy expenditure during walking, heightened risk for falls, injuries, pain, and reduced quality of life” requires a reference.

Line 51: The same applies to citations.

Methodology

Missing Citations: In the statement “Some studies in Parkinson’s have attempted to go beyond simple nominal classification and assess changes in overall gait patterns in a more continuous or ordinal manner,” citations are missing. Which studies?

Throughout the work, several statements lack citations.

Method

Trials: Following, each participant completed a series of trials involving rhythmic auditory. How many trials?

Table 1: A table footnote explaining all abbreviations should be included. For example: Max: maximum; min: minimum.

Sentence: Is this sentence correct? “El Segundo, CA, USA) attached along the lower legs, upper legs, and pelvis, connected to a smartphone via Bluetooth.”

Statistical Analysis: There should be a section on statistical analysis. What software was used, what assumptions were checked, what tests were used for each objective? What was the established significance level?

Consistency Model: Why was the consistency model used instead of the absolute agreement model? Why were absolute reliability (SEM) and the minimum detectable change (MDC) not calculated?

Reliability

ICC: To assess the reliability of the HMM-SM, we computed the intraclass correlation coefficient (ICC). Why were other reliability metrics not calculated?

Table 4: Add a table footnote explaining the abbreviations.

Discussion

I believe there is little contrast and comparison with other studies. I suggest improving the discussion to enhance the quality of your manuscript. Six to seven references seem insufficient.

Limitations

The limitations are not clearly stated.

I hope these comments are helpful for improving your manuscript

Author Response

Please see the attachment for itemized responses.

Reviewer 2 Report

Comments and Suggestions for Authors

In this paper, a novel approach is explored, which uses the similarity measure based on hidden Markov model (HMM-SM) to evaluate the change of gait pattern. The approach is based on the gyroscope and accelerometer signals from 1-2 inertial sensors. Eleven normal individuals were equipped with a system that disturbed gait patterns by manipulating station-time symmetry. A total of 11 healthy volunteers participated in the experiment. The experiment was conducted professionally, and the discussion was adequate. Overall, this paper could be published in Sensors journal after all the comments are well addressed.

1. For broad impact, some related works of flexible wearable sensors are suggested to be referred to in the introduction part, i.e., https://doi.org/10.1016/j.cej.2024.154445; https://doi.org/10.1002/adfm.202406789.

2. The higher ICC mentioned in the manuscript indicates that the output of the similar group is similar, and the standard is given, but Figure 4 only shows the test component and does not explicitly refer to the ICC value tested.

3. The experiment was carried out in the laboratory, and the participants only walked in a straight line. It is suggested that the author explore the performance of this method in more dynamic real environments ( such as outdoor walking, turning ).

4. HMM-SM showed good reliability for combined thigh signal (ICC-0.803) and calf signal (ICC-0.795), but for a single sensor, especially in the pelvic position, the reliability of HMM-SM was low (ICC = 0.59). It is suggested that the author further optimize the analysis ability of single sensor configuration.

5. When the article shows the results, the number of charts is limited, and the visualization of data is somewhat lacking. Although there are some tables showing the results of different sensor configurations, more histograms and line charts can be used to show the effects of different model parameters and configurations more intuitively. This can help readers understand the model performance more clearly.

Author Response

Please see the attachment for the itemized responses.

Reviewer 3 Report

Comments and Suggestions for Authors

This paper proposes a similarity measure of gait asymmetry based on HMM applied to time series of IMU sensors, aimed at the assessment of gait deficits or monitoring rehabilitation, reducing the need of instrumentation (using a small number of sensor units) and data for training the models (it is an unsupervised method, that is trained on a patient-specific basis).

The study conducted to test the validity and reliability of this method has two substudies: one to verify that the proposed measure varies in agreement with a reference asymmetry measure, and another to assess that it is sensitive and robust in the detection of changes.

The paper has a very good quality overall: the subject and purpose of the proposal and the study are well grounded and described in the introduction, the methods and results are clear, sound, detailed and reproducible, and there is a well substantiated discussion of the results, including a reasonable acknowledgement of its limitations.

My only major concern is about the segmentation of the data into gait cycles that precedes most of the analysis, as described in lines 168-171. If I understood correctly, the similarity measurement proposed by the authors was applied only to the “chunks” of the sensor signals corresponding to straight gait cycles extracted after that segmentation, which was done using the Xsens Awinda set of sensors. But the whole purpose of the proposed method is to reduce the complication of that instrumentation, and use a smaller set of sensors. So, how would the signals be segmented and identified as straight gait cycles without the Awinda set, in a “real-life” situation? Or how can the results be generalized in that situation, if such a segmentation cannot be done?

Another suggestion (rather than strong objection) is about the statistical analysis chosen to test the validity of the method compared to STSR (section 2.5.1, with results presented in 3.1). After a RM-ANOVA that modelled the expected outcomes of the similarity measurement (SM) depending on the underlying symmetry levels (SL) of STSR, the authors made post-hoc tests of paired differences, in order to verify that the differences between SM were approximately proportional to the differences between SL.

I like the general approach to this analysis (rather than e.g. trying to fit a linear regression between SM and SL, for the reasons that the authors themselves argued), but the amount of 8 post-hoc tests could have been reduced to 3, corresponding to the following hypotheses: (a) similarities between measurements of the same SL (SL1|SL1, SL2|SL2, and SL3|SL3) are equal; (b) similarities between measurements with “one step” between SL (SL1|SL2, and SL2|SL3) are also equal; and (c) the similarities decrease the larger the distance between the two SL.

I acknowledge, though, that those kind of comparisons need a bit of more advanced calculations than the commonplace pairwise comparisons (they could be done via linear hypotheses applied to statistical models — cf. John Fox’s R Companion to Applied Regression), and the analysis made by the authors is nevertheless correct, well described and explained, so it may be acceptable to leave it as it is.

Minor comments:

Line 78: The work cited as by “Sahraeian et al.” should be “Sahraeian and Yoon” (only two authors).

Line 180: “similarity of the trained HMMs compared to the expected similarity based on the mean symmetry levels”. I didn’t understand what that “expected similarity” was, until I reached section 2.5 later on, where Figure 2 explains the idea better. I suggest to reword this to avoid such doubts, perhaps just referring forward to that section or figure.

Line 278: “multiplied the raw p-values by the correction factor”. What factor? Was it Bonferroni’s correction, Holm, or another one?

Figure 3 is in the section of methods, but it actually mixes methods and results. Besides, the right plot (b) has many small details that cannot be clearly read.

Author Response

(The authors gave the same response as above.)
